# Molecular Dynamics of Atomic Layer Deposition: Sticking Coefficient Investigation

**Thokozane Justin Kunene** [1,*], **Lagouge Kwanda Tartibu** [1], **Sina Karimzadeh** [2], **Peter Ozaveshe Oviroh** [2], **Kingsley Ukoba** [2] **and Tien-Chien Jen** [2]

1   Department of Mechanical and Industrial Engineering Technology, University of Johannesburg, Johannesburg 2006, South Africa; tkunene@uj.ac.za, tkunene@uj.ac.za (T.J.K.); ltartibu@uj.ac.za (L.K.T.)
2   Department of Mechanical Engineering Science, University of Johannesburg, Johannesburg 2006, South Africa; skarimzadeh@uj.ac.za (S.K.); poviroh@uj.ac.za (P.O.O.); kukoba@uj.ac.za (K.U.); tjen@uj.ac.za (T.-C.J.)
*   Correspondence: tkunene@uj.ac.za; Tel.: +27-11-559-6978

**Abstract:** This study focused on the atomic scale growth dynamics of amorphous $Al_2O_3$ films microscale structural relaxation. Classical Molecular Dynamics (MD) can not entirely model the challenging ALD dynamics due to the large timescales. The all-atom approach has rules based on deposition actions modelled MD relaxations that form as input to attain a single ALD cycle. MD relaxations are used to create a realistic equilibrium surface. This approach is fitting to this study as the investigation of the sticking coefficient is only at the first monolayer that includes the layering of a hydroxyl surface of alumina. The study provides insight between atomic-level numerical information and experimental measurements of the sticking coefficient related to the atomic layer deposition. The MD modeling was for the deposition of $Al_2O_3$, using trimethylaluminum (TMA) and water as precursors. The film thickness of 1.7 Å yields an initial sticking coefficient of TMA to be $4.257 \times 10^{-3}$ determined from the slope of the leading front of the thickness profile at a substrate temperature of 573 K. This work adds to the knowledge of the kinetic nature of ALD at the atomic level. It provides quantitative information on the sticking coefficient during ALD.

**Keywords:** atomic layer deposition; film thickness; molecular dynamics; sticking coefficient

## 1. Introduction

Atomic Layer Deposition (ALD) is a technique in which precursor vapors are alternatively pulsed onto a substrate's surface, with an inert gas purge cycle in between. The surface reactions in ALD are complementary and self-limiting, allowing for material deposition through highly uniform and conformal development with atomic-level thickness control [1–4]. Classical molecular dynamics, which employs predetermined potentials and force fields based on empirical data or independent electronic structure computations, has long been recognized as a valuable method for studying a wide range of complex condensed matter systems, including biomolecular assemblies. In molecular dynamics, the typical approach is to calculate these potentials ahead of time. The entire interaction is typically broken down into two-body and many-body contributions, long-range and short-range terms, electrostatic and non-electrostatic interactions, all of which must be represented by appropriate functional forms [5–7].

There is an all-atom simulation of the ALD process that provides an atomic-level understanding of the ALD process. The algorithm in Figure 1 describes the deposition of consecutive precursor pulses in a physically reasonable manner. The simulation algorithm starts with a hydroxylated surface and looks for -OH groups on the surface, which are the active sites. After that, an accessible -OH is chosen at random for the ALD product deposition [8]. Hu et al. [8] proposed a different technique based on classical MD. Due to

the limited simulation durations for complex systems, MD alone cannot represent the whole ALD cycles. Hu et al. overcame this timescale limitation by linking the MD with a set of deposition criteria [9]. Deposition rules can be considered a function that considers an uncoated surface as input and a roughly coated surface as output. MD relaxation steps correct these approximations, resulting in a realistic surface structure fed into the next cycle's deposition rules [9].

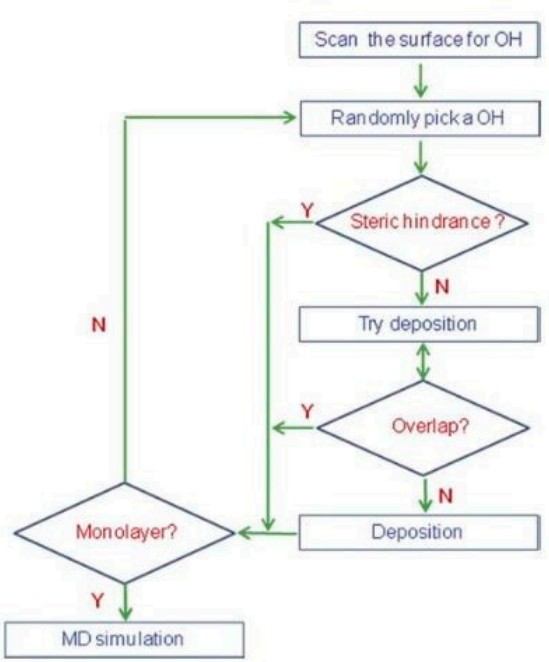

**Figure 1.** MD film growth deposition method [8]

The sticking coefficient (SC) is defined as the reaction probability, or the reactivity of a single precursor molecule with a reactive site on the surface. The SC is an intriguing metric since it may compare the efficiency of different ALD precursors and reactors [10]. Although the SC idea is extensively utilized, absolute quantities are rare in the literature [11]. The amount of time it takes to feed a saturated monolayer of precursor molecules is determined by the SC and the concentration of the precursor molecules. In general, feeding time increases as SC decreases [11]. Rose et al. [11] used the premise that desorption of chemisorbed precursor molecules is negligibly slow in ALD. The probability of adsorption of a single precursor molecule by a microscopic planar substrate can be calculated using Langmuir's theory (Surface chemistry [12]) of surface coverage [13,14]. The molecular symmetry of precursor molecules is an intriguing characteristic. Molecules with the same type of ligand have more symmetry than those with different types of ligands. If the SC of a single precursor molecule is dictated by the orientation of the precursor molecule during its interaction with the free reactive site, its value can be calculated from numerous surface interactions that are governed by the precursor molecule's symmetry. A large SC is caused by a high molecular symmetry, whereas a low molecular symmetry causes a small SC [13]. The initial sticking coefficient determines the maximum pumping speed per unit area that a given film can produce. The sorption capacity determines the time interval during which the film pumps at a given partial pressure. As a result, reliable estimations of such films' time-dependent pumping and interaction features are required [15].

The sticking coefficient (SC) is essential for optimizing the ALD process in high topographical structures. It describes the probability of a precursor adsorbed at the device's surface when it collides [16]. The work of Arts et al. [17] established a method for directly extracting initial sticking probabilities/coefficient from thickness profiles recorded in constructions with a high aspect ratio. They demonstrated that at high-aspect-ratio structures,

the leading front of the thickness profile provides direct information on the reactant sticking probabilities. The front slope of the thickness was utilized to calculate the sticking probability of $Al(CH_3)_3$ and $H_2O$ during $Al_2O_3$ ALD. The initial sticking coefficient ($SC_0$) was giving $(0.5 - 2) \times 10^{-3}$ for $Al(CH_3)_3$ and $(0.8 - 2) \times 10^{-4}$ for $H_2O$ at $T_{set}$ = 275 °C ($T_{sub}$ ~ 220 °C). The initial sticking coefficient value of $H_2O$ was shown to be temperature-dependent, decreasing to $(1.5 - 2.3) \times 10^{-5}$ at 150 °C. The film thickness determines the sticking coefficient—the film's thickness profiles slope increases as the deposition temperature increases. The experimental profiles show that the SC increases with increasing deposition temperature since steeper slopes correspond to higher SCs [13]. While lateral-high-aspect-ratio trenches were utilized for their study, the aforementioned method can also be used for other 3D features [17].

Träskelin et al. [18] investigated the sticking process of methyl radicals on carbon dangling bonds using molecular dynamics simulations. They discovered that the proximity of unsaturated carbon atom sites significantly impacts the chemisorption of a $CH_3$ radical onto a dangling bond. The structure and dynamics of water molecules and the hydrogen bonding network, formed when water molecules interact with a solid crystalline substrate, were examined by Argyris et al. using MD [19]. At room temperature, Argyris et al. [20] used molecular dynamics simulations to investigate the dynamic characteristics of water at the silica–liquid interface. They found that the anisotropic reorientation of water molecules is greatly influenced by the relative orientation of interfacial water molecules and their interactions with surface hydroxyl groups. The wide range of methodologies used to calculate the sticking coefficient like to use various experimental methods, such as Auger Electron Spectroscopy and QCM measurements, and theoretical approaches such as density functional theory (DFT) calculations or Monte Carlo modeling. More recently, sum-frequency generation has been reported in the literature [21].

This study aims to better understand the structure of precursor sticking coefficient during a high temperature of 573 K and high-pressure ALD process at a molecular level. The study focuses on the ALD deposition process of aluminum oxide, $Al_2O_3$, using trimethylaluminum (TMA—$(Al(CH_3)_3)$) and water ($H_2O$) as precursors. The SC has been determined from the thickness profiles generated from the MD approach of the ALD process. While lateral-high-aspect-ratio trenches were utilized in Arts et al.'s [17] study, we aim to prove that the same method could be used for other surface structure as it was used for our study.

## 2. Computational Details

### 2.1. The Simulation Approach

Hu et al. describes the complicated nature of the ALD growth. Because of the physicochemical mechanisms that regulate ALD growth, a simulation that can depict the time evolution of interacting atoms during the ALD process is critical. An empirical interatomic potential is a well-suited approach to describing ALD process and relaxation processes in MD simulations [8]. The study proposes two ALD, half-reactions equations. The equations are as follows [8]:

$$AlOH + Al(CH_3)_3 \rightarrow Al - O - Al(CH_3)_2 + CH_4 \tag{1}$$

$$Al - O - Al(CH_3)_2 + 2H_2O \rightarrow Al - O - Al(OH)_2 + 2CH_4 \tag{2}$$

The interatomic potentials for the hydroxylated alumina surface were determined by Matsui [22]. They are of the Buckingham type. The pair's interaction potential between atoms *i* and *j* as a function of the inter-atomic distance $r_{ij}$ is given by Equation (3) as:

$$U_{ij} = \frac{q_i q_j}{r_{ij}} + D\left(B_i + B_j\right)\exp\left(\frac{A_i + A_j - r_{ij}}{B_i + B_j}\right) - \frac{C_i C_j}{r_{ij}^6} \tag{3}$$

The repulsion and dispersion interactions are included in this potential, which comprises a long-ranged electrostatic potential and a short-ranged Buckingham potential. D (4.184 kJ⁻¹mol⁻¹) is the force constant, $r_{ij}$ is the relative distance between atoms $i$ and $j$, and q is the effective charge on each atom in the equation above. A, B, and C, the empirically fitted parameters, are provided in Table 1.

**Table 1.** DFT the inter-atomic potentials [9]

| Element | $q$(e) | A(Å) | B(Å) | $C\left(\text{Å}^{3}\text{ kJ}^{1/2}\text{ mol}^{-1/2}\right)$ |
|---------|--------|------|------|------|
| Al | 1.4175 | 0.78520 | 0.03400 | 36.82 |
| O | −0.9450 | 1.82150 | 0.13800 | 90.61 |
| H | 0.4725 | −0.17607 | −0.02462 | 1.86285 |

The molecular dynamics simulation was built in the LAMMPS-code (Large-scale Atomic/Molecular Massively Parallel Simulator) offered by SANDIA [23]. The visualization and post-processing of results were performed in OVITO (open visualization tool) [24]. In LAMMPS, the cut-off radius is specified by the potential. Short-range interactions are only taken into account in some MD implementations. All MD simulations were performed in the canonical ensemble with a time step of 0.1 fs (NVT). The number of particles (N), the volume of the simulation box (V), and the temperature (T) were all kept constant [20]. The short-range Buckingham interactions were determined using a cut-off of 1 Å. The long-range Coulombic interaction cut-off was set at 50 Å, accommodating the changing periodic boundary. In all MD simulations, the Nosé–Hoover chain technique [25] was employed to integrate the equations of motion. A two-dimensional correction was made to the Ewald summation below $1 \times 10^{-7}$ for Coulombic forces. The number of computations is lowered by including neighbor lists, which track which atoms are predicted to interact on the next timestep. This strategy saves time by avoiding the need to calculate all-atom pair distances in the simulation at each timestep. Binning is considered to divide the simulation domain into smaller portions, as another alternative. After then, atoms need to look for interactions in their bin and neighboring bins within the cut-off radius [26]. At 1 Å above the surface, the atoms' mobility was stopped. As a result, any molecules attempting to escape the simulation box by breaking loose from the surface were repelled. Zero velocities were subjected to a setforce, fix-command of LAMMPS. The fix-command froze certain atoms in the simulation by reducing their force to zero throughout a specified range.

### 2.2. Substrate Preparation

As detailed in Hu et al. and Brown et al. [8,9], the initial alumina surface model was made by annealing an alumina model after randomly replacing Al atom. The ALD of $Al_2O_3$ was carried out on a hydroxylated $Al_2O_3$ surface. A bulk $Al_2O_3$ sample with 770-atoms was generated according to the $Al_2O_3$ crystal structure 15.3836208, 20.5114944 and 25.639368 Å with equal angles, $\alpha = \beta = \gamma = 55.282635^0$. The simulation box was then anisotropically scaled to produce a large cubic supercell with the simulation box lengths of a = b = c = 26 Å, see Figure 2.

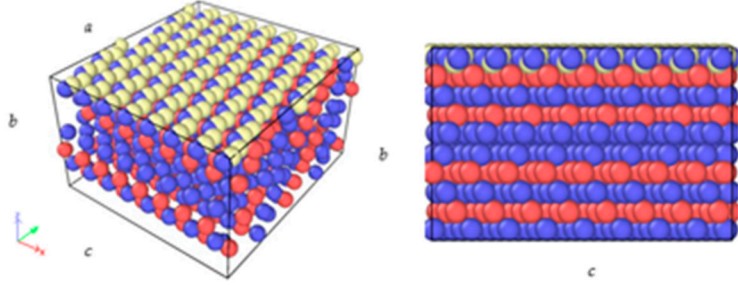

**Figure 2.** Al₂O₃ crystal structure with hydroxylated surface, atoms are red—aluminum, blue—oxygen and yellow—hydrogen, the simulation box lengths of a = b = c = 26 Å.

Following the rules of the ALD process and relaxation processes in MD simulations, this study also considered that the ALD reactions occur only on the surface's active -OH groups. It is assumed that the products of the metal precursor pulse are fully hydroxylated. Lastly, it was assumed that there are no impurities from the metal precursors left in the ALD films. The reverse reactions are negligible and fit with the algorithm approximately describes the stochastic deposition process [8,9].

ALD deposition rule was used in this study as described by Hu et al. A monolayer was built as shown in Figure 3a–e. The simulation box was resized to produce the necessary oxide density after annealing the sample at 5000 K for 300 ps. The sample was then quenched at a rate of 47 K/ps to 300 K in 100 ps. The box was then duplicated twice in the *x* and *y* directions (to increase the system size), and the surface was formed by expanding the box length in the *z*-direction to 51.0 Å. To produce the appropriate H content, specified H atoms were injected above the frozen layers to generate a hydroxylated surface. An Al atom was chosen randomly and swapped with an H atom to add to the system. Simultaneously, an O atom was chosen randomly and removed to maintain the system's overall charge neutrality. Finally, the surface was annealed for 500 ps at 1000 K. After annealing, the H atoms migrated to the surface region, resulting in a -OH terminated surface. The migration of -OH groups on the surface increases the ALD growth rate, allowing the surface to accommodate more precursor molecules to react on the surface [8,9]. The atoms in the bottom 3 Å of the simulation box were frozen at their relaxed geometries for additional simulation studies [8,27]. At that point, it is assumed that the projectile flux to the surface is sufficiently low; the sample surface can be assumed to be in equilibrium before the next hit. Since it is assumed that the duration between incoming projectiles is long enough not to affect each other, the same sample can be utilized for all simulations in this situation. This approximation can produce valuable results only if the model is in thermal equilibrium and a steady-state concerning layer growth [28].

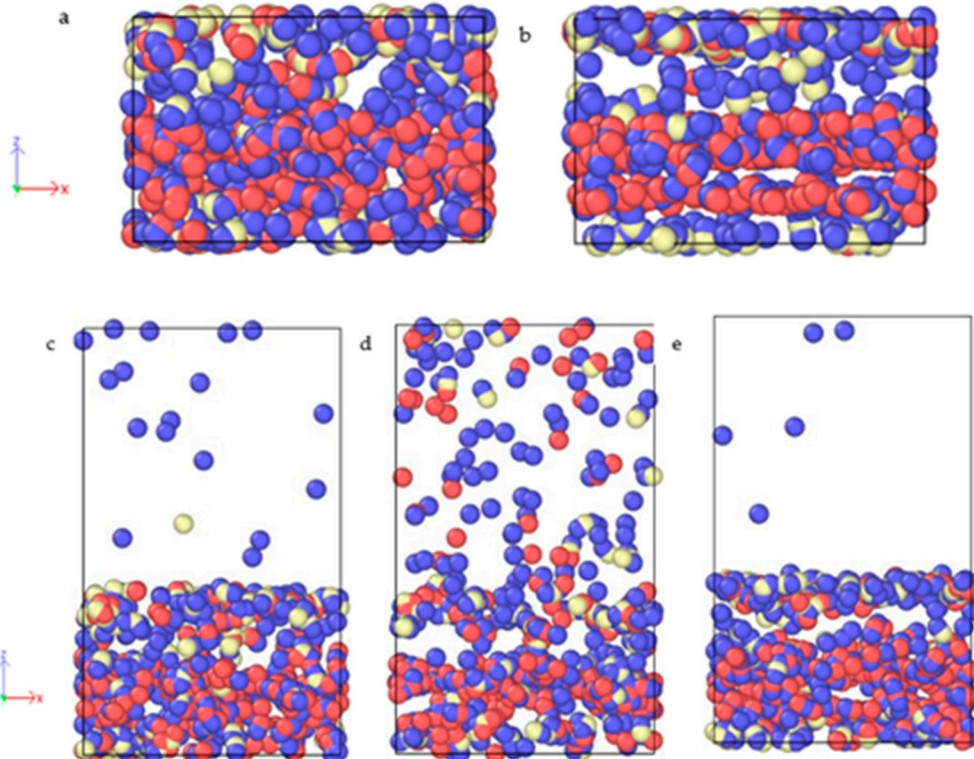

**Figure 3.** (**a**) After minimization (**b**) Annealing at 5000 K for 300 ps. (**c**) Quenched 300 K for 100 ps (**d**) Annealed at 1000 K for 500 ps (**e**) Relaxed hydroxylated surface at 300 K for 100 ps.

### 2.3. Model Validation

The predicted density reduction per cycle are plotted in Figure 4 as a function of the distance along the *z*-direction. The results are compared to those of Hu et al. [8], which studied the same physical system. As the number of deposited layers grows, both density profiles become flatter. This demonstrates that the ALD-deposited film has a lower density at the surface region in the early stages of ALD.

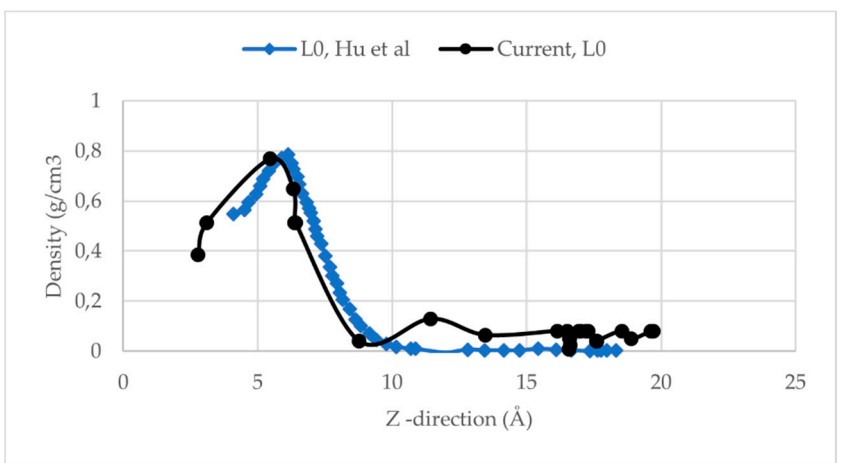

**Figure 4.** Model validation using a total density of the system at 300 °C at the first monolayer (1st ALD cycle).

### 3. Results

### 3.1. Al₂O₃ Film Thickness

Since a growth per cycle (GPC) is the amount of film thickness deposited in one cycle, the final thickness is determined by the cycle number rather than the response time [29]. The accurate prediction of the ALD response of the substrate is spatially dependent on the film thickness profile [30]. The precursor coverage $\theta$ is given by Equation (4) [14] as:

$$\Theta_{Al_2O_3}(x) = \frac{d_{ALD}(x)}{N_{cycles} \cdot GPC_{max}} \tag{4}$$

where $d_{ALD}$ is the ALD layer thickness, $N_{cycles}$ is the number of ALD cycles, and $GPC_{max}$ at an assumed full saturation level. The GPC found in this MD study is 1.2 Å that yields to a film thickness of 1.7Å (First cycle), where a GPC rate of 0.11nm (1.1 Å) per ALD processing cycle has been measured experimentally [30]. The precursor coverage $\theta$ was equivalent to 1. The surface was assumed to be fully covered. Figure 5a shows that to reduce the total energy of H-terminated Al₂O₃ surfaces, the atoms in the surface region rearrange to form a different structure than bulk Al₂O₃ to reduce repulsive interactions (the short-ranged Buckingham term in Equation (3)) and increase attractive electrostatic interactions between Al and O [7,8,31,32]. In Figure 5b, the temperature of the system is relaxed at 573 K. It shows that as the system's temperature rises, the kinetic energy of the system rises, and the frequency of the atoms' vibrations rises with it. The thermal motion becomes more vigorous as the temperature rises from a thermodynamic standpoint [33].

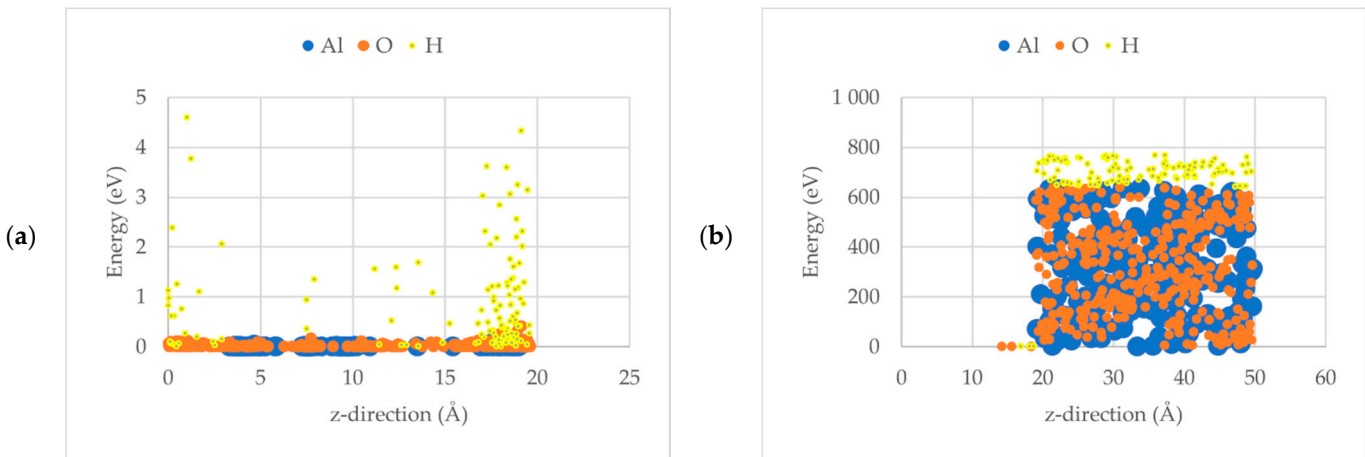

**Figure 5.** (**a**) Energy minimization of the system and (**b**) energy distribution after the first cycle along the growth length (*z*-direction).

Figure 6a,b visually demonstrate the layering of hydroxyls that are found in an ALD process. Most ALD procedures, for example, produce a sub-monolayer of growth after each cycle, which is usually attributed to steric effects from the precursor's ligands obstructing active sites, competing chemisorption pathways, or other factors [34]. It is observed that rather than being integrated into the majority of the film, the H atoms prefer to rise to the surface. Due to steric hindrance effects, the -OH distribution on the surface has a significant impact on the ALD growth mode. This suggests that avoiding populated -OH during the early stages of ALD is crucial for minimizing nucleation times.

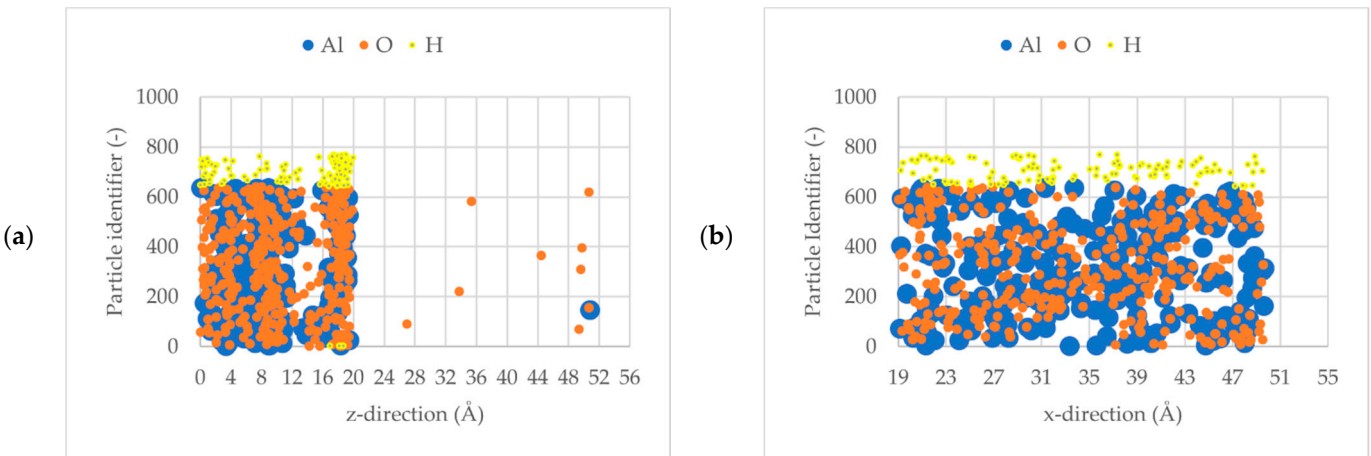

**Figure 6.** Surface after relaxation indicating -OH surface hydroxyls (**a**) -OH species attach to the surface of Al and strengthen its coordination to it (**b**) surface coverage across the substrate along the *x*-direction.

### 3.2. Initial Sticking Coefficient, So

Equation (5) was used to compute the initial sticking coefficient from a half coverage length of the substrate [17]. The initial sticking coefficient ($S_0$) governs the slope of the leading front of the thickness profile. The slope was found to be 0.035. It yields an initial sticking coefficient of $4.257 \times 10^{-3}$. The relation at the slop normalized to be $\tilde{z} = z / L_x$ (1D, one dimensional approach) at the point of half coverage $\theta = \frac{1}{2}$ along the *x*-direction, Equation (6).

$$\left|\frac{\partial \theta}{\partial \tilde{z}}\right|_{\theta=1/2} \approx \sqrt{s_0/13.9} \tag{5}$$

$$\left|\frac{\partial \theta}{\partial \tilde{x}}\right|_{\theta=1/2} \approx \sqrt{s_0/13.9} \tag{6}$$

The initial sticking coefficients of TMA found from other methods and substrate structures are listed in Table 2. This study found an initial sticking coefficient $S_0 = 4.257 \times 10^{-3}$ that fits in the range of $(2–7) \times 10^{-3}$ found by Arts et al. [17]. The method using a slope at a half the surface coverage is the simplest way of computing the sticking coefficient. Therefore, the $S_0$ found using an MD relaxation approach fits well with the same method from the literature like the high-aspect-ratio of trenches that was utilized by Arts et al. [17].

**Table 2.** Initial sticking coefficient of TMA during ALD of $Al_2O_3$, determined from the thickness profiles at 300 °C substrate temperature.

| Method | Researchers | $S_0$ |
| --- | --- | --- |
| Diffusion model, thickness profiles, lateral high-aspect-ratio structure (LHAR) | Ylilammi et al., Ref. [35] | $5.27 \times 10^{-3}$ |
| 1D structures, thickness profiles, lateral high-aspect-ratio structure (LHAR) | Arts et al., Ref. [17] | $(2–7) \times 10^{-3}$ |
| LHAR, thickness profiles, 3D structures, experiments and Monte Carlo simulations | Schwille et al., Ref. [14] | $2 \times 10^{-2}$ |
| LHAR, thickness profiles, 3D structures | Yim et al., Ref. [36] | $4 \times 10^{-3}$ |
| Molecular Dynamics (All-atom approach), thickness profiles, as Arts et al. Ref (Arts et al., 2019) | Current study | $4.257 \times 10^{-3}$ |

### 4. Conclusions

The key strength of this study was its focus on the atomic scale growth dynamics of amorphous $Al_2O_3$ films by inspecting microscale (MD) structural relaxation. The current findings add to a growing body of literature on applying the all-atom approach. There are rules based on the deposition actions to model MD relaxations of a single ALD cycle. The study supplemented the MD relaxations to create a realistic equilibrium surface by including the layering of hydroxyls on alumina. It was from a relaxed surface that an examination of the initial sticking coefficient of TMA was conducted. The study has gone some way towards enhancing understanding of the initial sticking coefficient between atomic-level numerical information and its experimental measurements related to the atomic layer deposition.

Therefore, the initial sticking coefficient was determined to be $4.257 \times 10^{-3}$, including error estimates from the MD model that would need improved interatomic potentials by using density functional theory (DFT) to account for the ligand exchanges in the presence of a hydroxylated surface. Notwithstanding these limitations, the study suggests that the initial sticking coefficient is an essential parameter for predicting the conformality of ALD growth. The ALD growth is calculated from the slope of the saturation profile. The saturation profile of ALD $Al_2O_3$ has a near-ideal form, namely, a constant thickness of 1.7 Å. This thickness develops from the first cycle, as expected for an ALD process. The process is based on repeated self-terminating and the gas–solid interactions as the surface coverage increases. Therefore, significantly beyond 50% coverage, the overall sticking coefficient decreases. This is because of the site-blocking hindering diffusion.

The consequence of unavoidable -OH groups that will always be present on the surface is supported by the current findings. Since viewed on the microscale, the most prominent finding from this study is that the -OH groups influence the sticking coefficient. Nevertheless, if the desorption of ligands is encouraged by increasing the surface temperature (to the limit of the TMA precursor, but observed in this study at 573 K), the adsorbed molecules will reduce steric hindrance. As also shown in this study, the steric hindrance

effects due to the -OH distribution on the surface significantly impact the ALD growth mode. The desired effect is that the surface needs to saturate quickly when the reactant diffuses to have a sharp profile front that will rapidly increase growth at fewer precursor doses but gain a steeper slope on the thickness. Therefore, avoiding populated -OH during the early stages of ALD assists with the rapid thickness growth. However, -OH groups at the surface lead to decreased sticking coefficient as they limit diffusion.

This study has shown that the initial sticking coefficient's influence promotes higher reactivity on the surface. The effect is best achieved from a stepper slope, which means the initial sticking coefficient of the TMA precursor will be higher $> 10^{-3}$.

This work adds to the knowledge of the kinetic nature of ALD at the atomic level. It provides quantitative information on the sticking coefficient during ALD. It shows the advantage of determining sticking coefficients from ALD thickness profiles without further modeling. Recent advancements in computing technology have allowed computational techniques to explore the physical interactions and chemical dynamics of ALD. Future studies should emphasize the ab initio molecular dynamic techniques for an entire ALD process.

**Author Contributions:** Conceptualization and methodology, T.J.K.; validation, S.K. and P.O.O.; formal analysis, T.J.K.; writing—original draft preparation, T.J.K.; writing—review and editing, K.U.; supervision, L.K.T.; funding acquisition, T.-C.J. All authors have read and agreed to the published version of the manuscript.

**Funding:** This research was funded by University of Johannesburg and the CSIR-DST Inter-Programme for funding the study.

**Institutional Review Board Statement:** Not applicable.

**Informed Consent Statement:** Not applicable.

**Data Availability Statement:** The data generated or analyzed during this study are available from the corresponding author on reasonable request.

**Acknowledgments:** The authors acknowledge the help from Mechanical and Industrial Engineering Department of the University of Johannesburg, JenNANO research group. A special acknowledgement is made to the Centre for High-Performance Computing (CHPC) for allowing us to use its resources to undergo this study.

**Conflicts of Interest:** There are no conflicts of interest declared by the authors. The funders had no involvement in the study's design, data collection, analysis, interpretation, manuscript writing, or publication of the findings.

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
