# Peer review of "Molecular Dynamics of Atomic Layer Deposition: Sticking Coefficient Investigation"

_applsci, doi:10.3390/app12042188_

Round 1
Reviewer 1 Report
In this manuscript the authors reported the investigation of aluminum oxide (Al2O3) film structure and composition by all-atom molecular dynamics (MD) simulations. Some interesting results are obtained. the atomic scale growth dynamics of Al2O3 films affected by small timescale structural relaxation. I therefore recommend an acceptance for publishing after next revisions.
1. Pages 1, abstract part, some background sentences can be added; The significance of this research should be emphasized;
2. Introduction part, if possible, some important and relative reports about self-assembled film nanostructures from various styles (Optics & Laser Technology 148 (2022) 107765; ACS Sustainable Chem. Eng. 2019, 7(12): 10888-10899.; ACS Sustainable Chem. Eng., 2020, 8(11): 4521-4536.) should be added to show clear background;
3. Fig. 3, please show clear image of packing after minimization or annealing;
4. table 2, please add more comparisons about initial sticking coefficient in recent other reports with similar systems;
5. Some minor Language error and style should be modified;
6. Fig. 6 missing description.
Author Response
Good day.
I have attached in this portal my response to your review comments. I have updated my manuscript accordingly as well.
Kind regards.

Reviewer 2 Report
Dear Authors,
This study needs some revision as follows:
The abstract and introduction part needs some improvement.
The structure of the introduction part needs to be rethought Why there are 1.1. Subsection in the introduction?
In the Introduction reference is made to Figure 1, and this is presented in part 2.
How does substrate temperature influence the SC?
After annealing, the H atoms migrated to the surface region, resulting in a -OH terminated surface.
How does the -OH groups the nature of the substrate surface influence?
The conclusion section needs significant changes. The findings should be shown clearly according to the objectives.
Author Response

(The authors gave the same response as above.)

Reviewer 3 Report
The theoretical investigation of aluminum oxide (Al2O3) film structure and compositions grown in the atomic layer deposition (ALD) using the molecular dynamics (MD) simulations is interesting and worth pursuing. The authors studied the atomic scale growth dynamics of amorphous Al2O3 films by separating large 12 timescale surface interactions from small timescale structural relaxation and simulated the growth of Al2O3 using trimethylaluminum (TMA) and water as precursors. The manuscript was well written and the simulation data and results were well discussed in the manuscript.
Some typing errors appear in the text, for example, in Eq. (1), there is “O’ in the right, but no “O” in the left. Please check the manuscript carefully and correct the errors.
Author Response

(The authors gave the same response as above.)

Round 2
Reviewer 2 Report
Dear Editor,
I recommend an acceptance for publishing in this form.
Best regards,
Cristina Cazan